# The Pelagic In situ Observation System (PELAGIOS) to reveal
# biodiversity, behavior and ecology of elusive oceanic fauna
Hoving, Henk-Jan[1], Christiansen, Svenja[2], Fabrizius, Eduard[1], Hauss, Helena[1], Kiko, Rainer[1],
Linke, Peter[1], Neitzel, Philipp[1], Piatkowski, Uwe[1], Körtzinger, Arne[1,3]
[1]GEOMAR, Helmholtz Centre for Ocean Research Kiel, Düsternbrooker Weg 20, 24105 Kiel, Germany.
[2]University of Oslo, Blindernveien 31, 0371 Oslo, Norway
[3]Christian Albrecht University Kiel, Christian-Albrechts-Platz 4, 24118 Kiel, Germany
Corresponding author: hhoving@geomar.de

## 1. Abstract

There is a need for cost-efficient tools to explore deep ocean ecosystems to collect baseline biological observations on pelagic fauna (zooplankton and nekton) and establish the vertical ecological zonation in the deep sea. The Pelagic In situ Observation System (PELAGIOS) is a 3000 m-rated slowly (0.5 m/s) towed camera system with LED illumination, an integrated oceanographic sensor set (CTD-$O_2$) and telemetry allowing for online data acquisition and video inspection (Low Definition). The High Definition video is stored on the camera and later annotated using annotation software and related to concomitantly recorded environmental data. The PELAGIOS is particularly suitable for open ocean observations of gelatinous fauna, which is notoriously undersampled by nets and/or destroyed by fixatives. In addition to counts, diversity and distribution data as a function of depth and environmental conditions (T, S, $O_2$), in situ observations of behavior, orientation and species interactions are collected. Here, we present an overview of the technical setup of the PELAGIOS as well as example observations and analyses from the eastern tropical North Atlantic. Comparisons to MOCNESS net sampling and data from the Underwater Vision Profiler are provided and discussed.

## 2. Introduction

The open ocean pelagic zones include the largest, yet least explored habitats on the planet (Robison, 2004; Webb et al., 2010; Ramirez-Llodra et al., 2010). Since the first oceanographic expeditions, oceanic communities of macrozooplankton and micronekton have been sampled using nets (Wiebe and Benfield, 2003). Such sampling has revealed a community typically consisting of crustaceans, cephalopods, fishes and some sturdy and commonly found gelatinous fauna (Benfield et al., 1996). Underwater observations in the open ocean via SCUBA diving

(Hamner et al., 1975) and later via submersibles (Robison, 1983; Robison and Wishner, 1990) and
in situ camera systems (Biard et al., 2016, Picheral et al., 2010) revealed that a variety of organisms
are much more abundant in the open ocean than previously estimated from net sampling (Robison,
2004).This was particularly true for fragile gelatinous zooplankton, a diverse taxonomic group of
different phyla, including the ctenophores and medusae (Remsen et al., 2004; Haddock, 2004) as
well as polychaetes (Christiansen et al., 2018), rhizaria (Biard et al., 2016) and pelagic tunicates
(Remsen et al., 2004; Neitzel, 2017) , which often are too delicate to be quantified using nets as
they are damaged beyond identification, or they are easily destroyed by the use of common
fixatives.
Underwater (*in situ*) observations in the pelagic ocean not only revealed a previously unknown
community, they also allowed the collection of fine-scale distribution patterns in relation to biotic
and abiotic factors (e.g. Haslob et al., 2009; Möller et al., 2013; Hauss et al., 2016) as well as
information on posture, interactions, and behavior (Hamner and Robison, 1992; Robison, 2004;
Robison, 1999; Hoving et al., 2017). Submersibles have proven to be valuable instruments to study
deep-sea pelagic biology (e.g. Robison, 1987; Bush et al., 2007; Hoving et al., 2013; 2016). Using
video transecting methodology, pelagic ROV surveys have been applied to study inter and intra-
annual variation in mesopelagic zooplankton communities (Robison et al., 1998; Hull et al., 2011)
and to explore deep pelagic communities in different oceans (Youngbluth et al., 2008; Hosia et al.,
2017; Robison et al., 2010). However, due to high costs as well as technological and logistical
challenges, regular submersible operations are still restricted to very few institutes and
geographical locations. Hence, there is a need for the development of additional more cost-
effective methodologies to explore and document deep-sea communities via in situ observations.

In the last decades, a variety of optical instruments has been developed to image and quantify plankton *in situ* (Benfield et al., 2007). The factors that typically differentiate the available plankton imaging technologies are the size fraction of the observed organisms, illumination type, resolution of collected images/video, depth rating, deployment mode (e.g., autonomous, towed, CTD-mounted) and towing speed. Examples of instruments include the autonomous Underwater Vision Profiler (UVP5; Picheral et al., 2010), the Lightframe On-sight Key species Investigations (LOKI; Schulz et al., 2009) and towed plankton recorders (ISiiS; Cowen and Guigand 2008; for review see Benfield et al., 2007). These instruments can be deployed from ships of opportunity and collect detailed information on fine-scale distribution and diversity patterns of particles and plankton. The data reveal biological patterns on a global scale (Kiko et al., 2017) and of previously underappreciated plankton species (Biard et al., 2016). More recently, optical (and acoustic) instruments have been combined with autonomous gliders, rapidly increasing spatial resolution (Ohman et al. 2019).

Various towed camera platforms have been developed that can obtain video transect observations above the deep sea floor. Examples are the TowCam (WHOI), the DTIS (Deep Towed Imaging system, NIWA), the WASP vehicle (Wide Angle Seafloor Photography), OFOS (Ocean Floor Observation System, GEOMAR), and the more recent version OFOBS (Ocean Floor Observation and Bathymetry System; Purser et al., 2018). All these instruments are used for video or photo transects of the seafloor, with a downward looking camera, and typically a set of lasers for size reference. However, published descriptions of optical systems, other than ROVs and submersibles, that visualize macrozooplankton and micronekton (>1 cm) in the water column undisturbed by a filtering device or cuvette are, to the best of our knowledge, restricted to one (Madin et al., 2006). The Large Area Plankton Imaging System (LAPIS) is the only towed system that was developed

for the documentation of larger organisms in the water column (Madin et al., 2006). LAPIS
visualizes organisms between 1 and 100 cm, it combines a high-resolution color digital CCD
camera using progressive scanning interline-transfer technology with flashing strobes, and it is
towed at 1 knot via a fibre optic wire. LAPIS collects still images, illumination is sideways, and
organisms have to enter an illuminated volume to be visualized. Deployments in the Southern
Ocean enabled the reconstruction of depth distributions of the pelagic fauna (salps, medusae) but
also allowed some behavior observations, e.g. the moulting of krill (Madin et al., 2006). More
publications of data collected with LAPIS are unavailable to our knowledge. Other than LAPIS,
we wanted to develop a towed pelagic observation system that collects video during horizontal
transects (with forward projected light), in a similar way as pelagic ROV video transects, in order
to document behaviour in addition to diversity, species-specific distribution and abundance data
of pelagic fauna.
The functional requirements for the instrument were the ability to: (1) visualize organisms > 1 cm
in waters down to 1000 m with high-definition video, (2) deploy the instrument from ships of
opportunity in an autonomous or transmitting mode, (3) make it lightweight and practical so it can
be deployed easily and safe with two deck persons and a winch operator, (4) enable correlation of
observations with environmental parameters (S, T, $O_2$) and other sensor data, and (5) make
observations comparable to ROV video transects in other reference areas. We present a description
of the Pelagic In situ Observation System (PELAGIOS), examples of the kind of biological
information it may gather, as well as biological discoveries that have resulted from deployments
on research cruises in the eastern tropical North Atlantic.

## 3. Pelagic In Situ Observation System

### 3.1 Technical Specifications

The PELAGIOS consists of an aluminum frame (length = 2 m) that carries the oceanographic equipment (Figure 1). White light LED arrays (4 LEDs produced at GEOMAR, 2 LED arrays (type LightSphere of Deep-Sea Power and Light ©) which illuminate the water in front of the system are mounted on an aluminum ring (diameter = 1.2 m). Power is provided by two lithium batteries (24V; 32 Ah) in a deep-sea housing. High-definition video is collected continuously by a forward viewing deep-sea camera (type 1Cam Alpha, SubC Imaging ©) which is mounted in the center of the ring. We used the maximum frame rate of 50 frames $s^{-1}$ but a lower frame rate is possible. A CTD (SBE 19 SeaCAT, Sea-Bird Scientific ©) with an oxygen sensor (SBE 43, Sea-Bird Scientific ©) records environmental data. A deep-sea telemetry (DST-6, Sea and Sun Technology ©; Linke et al., 2015) transmits video and CTD data to a deck unit on board allowing a low-resolution preview (600 x 480 lines) of the high definition video that is stored locally on the SD card (256 GB) of the camera. The power from the batteries is distributed to the LEDs via the camera. The 1Cam Alpha camera is programmable in such a way that there is a delay between providing power to the camera (by connecting to the battery) and the start of recording and switching on the LEDs. This enables the illumination to be turned on only underwater, and prevents overheating of the LED arrays while out of the water. During a cruise with the German research vessel MARIA S. MERIAN (MSM 49) we mounted a steel scale bar in front of the camera at a distance of 1 m. The distance between the centers of the white marks on the bar measured 5 cm.

### 3.2 Video transects

The PELAGIOS is towed horizontally at specified depths of 20-1000 m. The standard towing speed over ground is 1 knot (0.5 m/s), and the speed is monitored via the ship's navigational

system. A video transect at a particular depth can take as long as desired and is terminated by
lowering the PELAGIOS to the next desired depth. Maximum deployment time with full batteries
is approximately 6 hours.  The typical transect duration is 10-30 min. The depth of the PELAGIOS
can be monitored via online CTD data. Figure 2 shows the trajectories of the PELAGIOS at
different depths in the water column during a video transect down to 700 m. The deployment from
deck into the water and the reverse is fast and typically takes only about 5 min (see video clip in
the ESM. It is possible to deploy PELAGIOS in 'blind mode', where only the depth is monitored
using an online depth sensor (e.g., Hydrobios ©) and the video (without transmitted preview) is
recorded locally on the camera. The system can be operated completely blind (i.e., with no
communication between deck and underwater unit) where the target depth is estimated from the
length and angle of the wire put out, and the actual depth is recorded on the system by CTD or an
offline pressure sensor e.g. SBE Microcat ©.

**3.3 Video analysis and curation**
After a deployment, the video (consisting of individual clips of one hour) is downloaded from the
camera. Synchronisation between video and CTD data is done by setting all instruments to UTC
prior to deployment, which allows the data and video to be linked during analysis. The video is
annotated using the Video Annotation and Reference System VARS developed at the Monterey
Bay Aquarium Research Institute (Schlining and Jacobsen, 2006). This annotation program allows
for frame grabs from the video including time code. A Knowledge Base allows for inserting
taxonomic names and hierarchy, and a Query allows for searching the created database. While
many kinds of annotation software are available (for review see Gomes-Pereira et al., 2016), we
consider VARS the most suitable for our purposes since it combines the features of high resolution
video playback with a user friendly annotation-interface and the automatic creation of an
annotation database which can easily be accessed through the various search-functions and tools
of the Query. The taxonomic hierarchy and phylogenetic trees in the database are directly
applicable to our video transects. Since this software was developed by MBARI, which also
maintains the most extensive databases of deep pelagic observations, it makes communication
about and comparison of observations and data practical. Videos are transported on hard drives
after an expedition  and are transferred for long term storage on servers maintained by the central
data and computing centre at GEOMAR, providing instant access to videos and images with
metadata description via the media server ProxSys.

**3.4 Sample volume**
To estimate the sample volume of the PELAGIOS we compared video counts from the PELAGIOS
with concomitantly obtained abundance data from an Underwater Vision Profiler (UVP5; Picheral
et al., 2010). Four deployments from the R/V Maria S. Merian cruise MSM 49 (28.11.- 21.12.2015,
Las Palmas de Gran Canaria/Spain – Mindelo/Cape Verde) were used for the comparison where a
UVP5 was mounted underneath the PELAGIOS. The UVP5 takes between 6-11 images per second
of a defined volume (1.03 L) and thus enables a quantitative assessment of particle and
zooplankton abundances. Objects with an equivalent spherical diameter (ESD) >0.5 mm are saved
as images, which can be classified into different zooplankton, phytoplankton and particle
categories. For the comparison between PELAGIOS and the UVP5, we used the pelagic
polychaete *Poeobius* sp., as 1) this organism could be observed well on both instruments, 2)
*Poeobius* sp. is not an active swimmer and lacks an escape response and 3) it was locally very
abundant, thus providing a good basis for the direct instrument comparison.
The UVP5 images were classified as described in Christiansen et al. (2018). *Poeobius* sp.
abundance (ind m$^{-3}$) was calculated for 20 s time bins and all bins of one distinct depth step (with
durations of 10-11 minutes at depths <= 50 m, 19-22 minutes at depths < 350 m and 9-11 minutes
at depths >= 350 m) averaged. These mean abundances were compared to the PELAGIOS counts
(ind s$^{-1}$) of the same depth step. A linear model between the PELAGIOS counts as a function of
UVP5 abundance provided a highly significant relationship (linear regression: $p < 0.001$, *adjusted*
$r^2 = 0.69$; Figure 3). The linear regression slope $b$ (0.116 m$^3$ s$^{-1}$, standard error 0.01 m$^3$ s$^{-1}$) between
the PELAGIOS-based count ($C_{PELAGIOS}$, ind s$^{-1}$) and mean UVP-based abundance ($A_{UVP}$, ind m$^{-3}$):
$$C_{PELAGIOS} = b * A_{UVP} + a \quad \text{(Equation 1 )}$$

was used to estimate the volume recorded per time in m$^3$ s$^{-1}$ ($b$) and the field of view in m$^2$
($b$/towing speed) recorded by PELAGIOS.
From this calculation it can be derived that PELAGIOS recorded an average volume of 0.116 m$^3$ s$^{-1}$
at a towing speed of 1 knot (= 0.5144 m s$^{-1}$). A cross-sectional view field of approximately 0.23
m$^2$ of PELAGIOS can be expected, compared to a theoretical field of view (FOV) of 0.45 m$^2$ based
upon the maximum image dimensions (0.80 m * 0.56 m) at 1 m distance from the lens. We can
now calculate the individuals observed by PELAGIOS per time to individuals per volume. To do
so we use the number of individuals in one transect and divide this number by the duration of the
transect to obtain individuals/minute, and divide this by 60 to get the individuals/second. From the
UVP-PELAGIOS comparison we derived a conversion factor of 6 to calculate the number of
individuals per second to number of individuals per m$^3$. This value is then multiplied by the
conversion factor 6, and again multiplied by 1000 to go from m$^3$ to 1000 m$^3$.

**3.5 Abundance, size and diversity at an example station "Senghor NW"**

To provide an example of the type of data that can be obtained with the PELAGIOS, we report here on day and night video transects down to 950 m in the Eastern Tropical North Atlantic, on the northwestern slope of Senghor Seamount (17°14.2'N, 22°00.7'W; bottom depth of approximately 1000 m). The results from the video annotations show that faunal abundances depend on the depth of deployment, and time of the day. During two transects of 11 minutes at 400 m, 226 individuals (1066 Ind/1000m³) were encountered during the day (the three dominant organism groups were fish, euphausiids and appendicularians) compared to 196 individuals (591 Ind/1000m³) during the night (the four dominant organism groups are fish, chaetognaths, medusae and ctenophores). Overall abundance of chaetognaths, decapods and mysids, and somewhat for fishes was higher during the night. The peak of euphausiids' abundance at 400 m shifts to the surface at night (Figure 4). The higher abundance of decapods, mysids and chaetognaths at night may indicate lateral migration or daytime avoidance. The vertical migration that was observed for fishes and crustaceans was much less clear for the gelatinous zooplankton groups including medusae and appendicularians (Figure 4). Ctenophores and siphonophores were abundant in the surface at night (but we did not perform transects at 20 and 50 m during the day) and the thaliaceans migrated vertically and were most abundant in shallow waters at night. The total number of annotated organisms for the daytime transects (total transect time 187 minutes; max. depth 950 m) was 835 compared to 1865 organisms for the longer nighttime transects (total transect time 292 minutes; max depth 900). Remarkable is the enormous abundance of gelatinous zooplankton (128) annotated organisms (899 Ind/1000m³) belonging to the three dominant groups of Ctenophora (53), Siphonophorae (21) and Thaliacea (44) in the topmost layer (20 m) at night. Below this layer, the depth profile shows a minimum in numbers of annotated individuals at 100, 200, and 300 m

water depth with a smaller peak of 57 gelatinous organisms (299 Ind/1000m³) in 450 m. Compared
to this, the depth distribution at day time shows a more regular, almost Gaussian shape with a
maximum of 31 (254 Ind/1000m³)  and 54 (254 Ind/1000m³) gelatinous organisms at 200 and 400
m water depth, respectively.

The faunal observations at station Senghor NW include a wide variety of taxa (Table 1; Figures 5
and 6), spanning in size from radiolarians to large siphonophores (such as *Praya dubia* and
*Apolemia*).Chaetognaths were the dominant faunal group. Typical examples of fragile organisms
that were not present or identifiable in the MOCNESS samples from the same cruise (Christiansen
et al 2016; Lüskow et al in prep.) but which can be efficiently observed by PELAGIOS include
large larvaceans (probably *Bathochordaeus* and *Mesochordaeus*), pelagic polychaetes (*Poeobius,*
*Tomopteris*) (Figure 5), and smaller siphonophores (such as *Bargmannia* and *Lilyopsi*; the latter
can be easily distinguished by their fluorescent body parts).(Figure 5). Observed medusae
belonged to the genera *Periphylla*, *Halitrephes*, *Haliscera*, *Crossota, Colobonaema*, *Solmissus* and
*Solmundella* (Figure 5). Venus girdles (*Cestum* spp.), *Beroe*, cydippids and lobate ctenophores
(such as *Thalassocalyce inconstans*, *Leucothea*, *Bathyceroe*, see Harbison et al., 1978 for
differences in robustness among ctenophores) were encountered at Senghor NW (Figure 5).
Cephalopod observations were rare but small individual cranchiid squids were observed in the
upper 50 m at night. Mastigoteuthid squids were observed with their mantle in a vertical orientation
and with extended tentacles in waters below 500 m. One large squid, *Taningia danae* was observed
during a transit between transecting depths. Other pelagic molluscs include the nudibranch
*Phylliroe* and different pteropod species. Observed fishes are snipe eels, hatchet fishes, lantern
fishes and *Cyclothone*. Fishes are among the dominant organisms encountered during PELAGIOS
transects but it is often impossible to identify fishes to species level from the video.

**3.6 Individual behavior**
In situ observations by PELAGIOS video may reveal direct observations on individual behavior.
Decapod shrimps were observed to release a blue or green bioluminescent cloud after performing
their tail flip as part of the escape response (Figure 6d). Potential reproductive behavior was
observed for two specimens of krill which were seen in a what could be a mating position, and
salps were observed to reproduce asexually by the release of salp oozoids (Figure 6c). Feeding
behaviors were observed for large prayid siphonophores and calycophoran siphonophores which
had their tentacles extended. *Poeobius* worms were observed with their mucus web deployed to
capture particulate matter (Christiansen et al., 2018) (Figure 6a). Narcomedusae of the genus
*Solmissus* were observed with their tentacles stretched up and down, which is a feeding posture
(Figure 5). In situ observations by the PELAGIOS also showed the natural body position of pelagic
organisms. Snipe eels were observed in a vertical position with their heads up, while dragonfishes
and some myctophids were observed in an oblique body position with their head down (Figure
6b).

**4. Discussion**
PELAGIOS is a pelagic ocean exploration tool that fills a gap in the array of observation
instruments that exist in biological oceanography, as transparent and fragile organisms (> 1 cm)
are up to now undersampled by both net-based and optical systems. The PELAGIOS video
transects are comparable to ROV video transects and can be obtained in a cost-effective way. The
resulting data can provide information on diversity, distribution and abundance of large (> 1cm),
fragile zooplankton and some nekton, and also of rare species. Due to the collection of HD color
video, behavior, color and position in the water column are documented which may provide
additional ecological information. Thus, the system complements gear that are suitable for
stratified observations and collections of robust mesozooplankton and micronekton (MOCNESS,
Hydrobios Multinet, and others) and optical systems that are suitable for high-resolution sampling
of small and abundant organisms (e.g. VPR, UVP5) (e.g. Benfield et al., 2007; Picheral et al.,
2010; Biard et al., 2015).  The instrument can be deployed with a small team and from vessels of
opportunity, in transmission or 'blind' mode. The relatively simple design limits technical failures
and makes the PELAGIOS a reliable tool for oceanic expeditions. While thus far the system has
only been deployed in the open ocean, it can be used in any pelagic environment with water that
has reasonable clearance and visibility. The data obtained after annotation of the video can be
uploaded into databases (e.g., the large database PANGAEA) after publication of the results
allowing for efficient data sharing and curation.
The clear distribution patterns that we observed in some animal groups (fish, crustaceans and some
gelatinous fauna) after annotating the video transects confirms that established biological
processes such as diurnal vertical migration (e.g. Barham, 1963) can be detected in PELAGIOS
data, and that the distribution data that we observe for encountered organisms are representative
for the natural situation. It has to be noted, though, that while the observed distribution patterns
should be representative, care must be taken with regards to abundance estimates of especially
actively- and fast-swimming organisms. Some fish and crustaceans react to the presence of
underwater instrumentation (e.g. Stoner et al., 2008). Gear avoidance (e.g. Kaartvedt et al., 2012)
can lead to an underestimation of abundance, whereas attraction to the camera lights (e.g. Utne-
Palm et al, 2018; Wiebe et al., 2004) would result in an overestimation. The large bioluminescent
squid *Taningia danae* seemed to be attracted to the lights of the PELAGIOS, and attraction
behaviour of this species has been described in other publications (Kubodera et al., 2007).
Compared to day transects, the high abundance of gelatinous organisms close to the surface during
night is likely to be partly an effect of the higher contrast in the videos of the night transects and
better visibility of the gelatinous fauna than during day transects. Therefore we did not perform
transects shallower than 50 m during the day. Many of the observed gelatinous fauna might be
present as well at shallow depths during day-light but are not detectable at 'blue-water-conditions'.
The difference between encountered taxa during the day and night transect may also be due to
trapping of organisms at the slopes of Senghor Seamount during the day (Isaacs and Schwartzlose,
1965; Genin, 2004) or by other causes for patchiness (Haury et al., 2000). However, from a
methodological side it should be noted that while the ship's towing speed is typically 1 knot, the
current speeds at the survey depths may differ, also between day and night. Currents may result in
more or less sampled volume of water and hence a variation in plankton being visualized. Since
abundance estimation relies on an accurate determination of the image volume, it needs to be
pointed out that it is our aim to better technically constrain the image area in future developments
(now derived from UVP quantitative observations) and to include flowmeter measurements.
After annotation, the PELAGIOS video transects may be used to reconstruct species-specific
distribution patterns, which can be related to environmental gradients (Neitzel, 2017; Hoving et
al. in prep.). Such data are valuable for overlap comparison in distribution patterns of consumers
and food items (see e.g. Haslob et al., 2009; Möller et al., 2012). The data can also be used in
biological studies that aim to predict the consequences of a changing ocean with altering
environmental gradients for species' distributions, as it has been done for net sampling of
mesozooplankton (Wishner et al., 2013) One example of changing environmental gradients is the
global trend of oxygen loss in the world oceans (Oschlies et al., 2018). Oxygen minimum zones
(OMZs) are occurring naturally in the mesopelagic zone (Robinson et al., 2010), and in different
oceans they have been found to expand horizontally and vertically as a result of climate change
(Stramma et al., 2008; Oschlies et al., 2018). Expansion of OMZs may result in a habitat reduction
of the pelagic fauna (e.g., Stramma et al., 2012), or increase the habitat for species with hypoxia
tolerance (Gilly et al., 2013). To predict the potential consequences of OMZ expansion for pelagic
invertebrates we investigated the abundance and distribution of distinct large gelatinous
zooplankton species, including medusae, ctenophores, siphonophores and appendicularians, in the
eastern tropical North Atlantic using PELAGIOS video transects and correlated the biological
patterns to the oxygen gradients (Neitzel, 2017; Hoving et al., in prep.).
During various cruises, the UVP5 was mounted underneath the PELAGIOS providing concomitant
data on macrozooplankton and nekton (PELAGIOS) as well as particles and mesozooplankton
(UVP5). The combination of the two instruments provides a great opportunity to assess both the
mesopelagic fauna and particles during one sampling event. The joint deployment of the
PELAGIOS and UVP5 also allowed an estimation of the sampled water volume of the PELAGIOS
as described above. The linear relationship between counts of the non-moving *Poeobius* sp. with
UVP5 and the PELAGIOS indicates comparability of the two different methods for animals in this
size class and provides a correction factor to estimate organism abundance (ind m$^{-3}$) from
PELAGIOS count (ind s$^{-1}$) data.
The field of view (FOV) derived from the UVP5 comparison for the PELAGIOS was estimated to
be 0.23 m$^2$ in comparison to 0.45 m$^2$ based on measurement of the scale bar at 1 m from the camera.
The angle of view of the PELAGIOS is 80° and therefore the field of view (FOV) is much smaller
than the FOV of video transects with a wide-angle lens e.g. by ROV Tiburon (Robison et al.,
2010). When comparing the FOV, it is important to take into account the object that is observed.
We provided an estimate of the FOV using *Poeobius* sp., which is a small organism that can be
detected only when it is close to the camera. Therefore, the area of the FOV for quantification of
*Poeobius* sp. is smaller than when quantifying larger organisms, and the initial identification
distance differs between species (Reisenbichler et al., 2017).
We compared PELAGIOS video transects with MOCNESS net (opening 1 m$^2$) abundance data by
integrating the PELAGIOS counts over the respective depth strata of the MOCNESS that happened
at the same cruise (Lüskow et al in prep.). The diversity of the gelatinous zooplankton in the total
MOCNESS catch is much lower (8 different taxa) (Lüskow et al., in prep.) than in the pooled video
transects (53 different annotated taxa) on the same station. The ctenophore *Beroe* is an example of
a gelatinous organism captured in MOCNESS hauls and also observed on PELAGIOS transects.
Normalization and subsequent standardization of the encountered *Beroe* in MOCNESS and
PELAGIOS transects show that on the same station and the same depths, PELAGIOS observes 3-
5 times more *Beroe* at the three depths where they were encountered by both instruments.
Additionally, the PELAGIOS also repeatedly observed *Beroe* at depths where they were not
captured by MOCNESS at all (although there were also depths where PELAGIOS did not observe
any *Beroe*). Preliminary comparisons of the data obtained with PELAGIOS and with MOCNESS
indicate substantial differences in the documented fauna, a phenomenon also observed in previous
comparisons between optical and net data (Remsen et al., 2004). Many more gelatinous taxa were
observed during PELAGIOS video transects than were captured in MOCNESS catches at the same
station (data presented here, Lüskow et al., in prep.) due to the delicate nature of many ctenophores,
medusae and siphonophores, preventing their intact capture by nets. A notable exception are the
small and robust calycophoran colonies of the families Diphyidae and Abylidae which were also
captured by MOCNESS. In contrast, avoidance behavior of strongly and fast swimming jellyfish
(e.g. *Atolla, Periphylla*), which may escape from the relatively slowly towed PELAGIOS, may
explain their increased occurrence in nets compared to video recordings. While PELAGIOS is
certainly suitable for visualizing delicate gelatinous fauna, it cannot replace net-sampling since
complementary specimen collections are needed to validate the identity of organisms that were
observed during PELAGIOS video observations. Therefore, it is desired that net tows with open
and closing nets such as Multinet Maxi or MOCNESS are performed in the same areas, or that
collections during submersible dives are made. An advantage of ROVs over PELAGIOS is the
ROV's ability to stop on organisms for detailed close up recording and potentially the collection
of the observed organisms. This is not possible with PELAGIOS as the ship is towing the
instrument.
While the imaging processing pipeline is not as streamlined as in other optical systems that use
still images such as the VPR or the UVP5, the potential of the PELAGIOS as an exploration tool
is illustrated by the discovery of previously undocumented animals. An example is the ctenophore
*Kiyohimea usagi* (Matsumoto and Robison, 1992) which was observed seven times by the
PELAGIOS and once by the manned submersible JAGO during cruises in the eastern tropical
North Atlantic. This large (>40 cm wide) lobate ctenophore was previously unknown from the
Atlantic Ocean and demonstrates how in situ observations in epipelagic waters can result in the
discovery of relatively large fauna (Hoving et al., 2018). Since gelatinous organisms are
increasingly recognized as vital players in the oceanic food web (Choy et al., 2017) and in the
biological carbon pump (Robison et al., 2005), in situ observations with tools like the PELAGIOS
can provide new important insights into the oceanic ecosystem and the carbon cycle. But small
gelatinous organisms may also have a large biogeochemical impact on their environment. This
was illustrated by the discovery of the pelagic polychaete *Poeobius* sp. during the PELAGIOS
video transects in the eastern tropical North Atlantic (Christiansen et al., 2018). The observations
of the PELAGIOS provided the first evidence for the occurrence of *Poeobius* sp. in the Atlantic
Ocean. During the R/V Meteor cruise M119, *Poeobius* was found to be extremely abundant in a
mesoscale eddy. Following this discovery, it was possible to reconstruct the horizontal and vertical
distribution of Atlantic *Poeobius* in great detail using an extensive database of the UVP5 (956
vertical CTD/UVP5 profiles) in the eastern tropical North Atlantic, and to establish that the high
local abundance of *Poeobius* was directly related to the presence of mesoscale eddies in which
they substantially intercepted the particle export flux to the deep sea (Christiansen et al., 2018;
Hauss et al., 2016).


Future effort should be focused on improving the assessment of the sample volume by integrating
technology that can quantify it (e.g. current meters, a stereo-camera setup or a laser-based system).
A stereo-camera set up would also allow for size measurements of the observed organisms, which
could be beneficial to estimate the biomass of the observed organisms from published size-to-
weight relationships. It might also be possible to obtain similar information based on structure-
from-motion approaches that proved successful in benthic video imaging (Burns et al., 2015). The
PELAGIOS system can also be a platform for other sensors. For example, the PELAGIOS was
used to mount and test the TuLUMIS multispectral camera (Liu et al., 2018). Future developments
include the preparation of the system for deployments down to 6000 m water depth. The integration
of acoustic sensors would be valuable to measure target strength of camera observed organisms,
to estimate gear avoidance or attraction and to estimate biomass and abundance of organisms
outside the field of view of the camera. We strongly encourage the use of complementary
instruments to tackle the relative importance of a wide range of organisms in the oceanic pelagic
ecosystem.


**Author contribution**

This instrument was designed, tested and applied by Henk-Jan Hoving and Eduard Fabrizius.
Rainer Kiko and Helena Hauss developed the idea of combining the PELAGIOS with the UVP5.
Philipp Neitzel and Svenja Christiansen analyzed the data in this manuscript in consultation with
Henk-Jan Hoving, Rainer Kiko and Helena Hauss. Arne Körtzinger, Uwe Piatkowski and Peter
Linke added valuable input to the further development of the instrument and its application
and/or the data interpretation. All authors contributed to writing the paper. All authors approved
the final submitted manuscript.

**Data availability**
The datasets generated and/or analysed during the current study will be available in the
PANGAEA repository: https://doi.pangaea.de/10.1594/PANGAEA.902241


## Acknowledgements

Our sincere gratitude goes to Ralf Schwarz, Sven Sturm, and other colleagues of GEOMAR's Technology and Logistics Centre as well as Svend Mees for their indispensable support in design and construction during the development of PELAGIOS. We want to thank the crew of the research vessels METEOR, MARIA S. MERIAN and POSEIDON for their excellent support during research expeditions, and Bernd Christiansen (University of Hamburg) for collaboration and leading of the expedition MSM49. Anneke Denda and Florian Lüskow are acknowledged for their help on the MOCNESS samples of gelatinous zooplankton collected during MSM49. Shiptime on RV Maria S. Merian and supporting funds were provided by the German Research Foundation (DFG) (grant MSM49 to Bernd Christiansen). We also thank the DFG for providing financial support to HJH under grants HO 5569/1-2 (Emmy Noether Junior Research Group) and a grant CP1218 of the Cluster of Excellence 80 "The Future Ocean". RK and SC were supported by grant CP1650 of the Cluster of Excellence 80 "The Future Ocean". "The Future Ocean" is funded within the framework of the Excellence Initiative by the DFG on behalf of the German federal and state governments. RK and HH were supported by the DFG as part of the Collaborative Research Centre (SFB) 754 "Climate-Biogeochemistry Interactions in the Tropical Ocean".

436

437

438

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

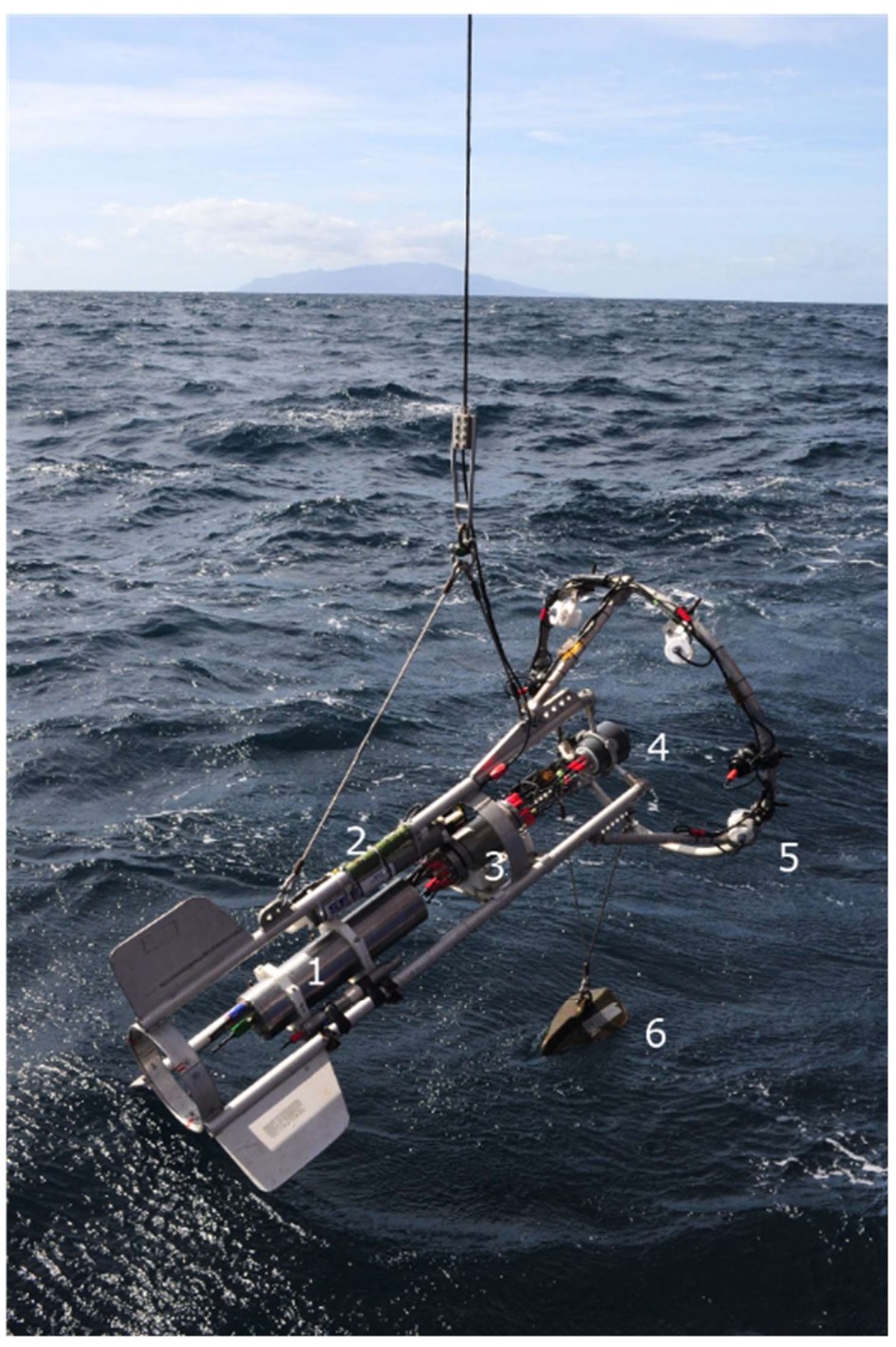

Figure 1: a) The Pelagic In Situ Observations System (PELAGIOS) with battery (1), CTD (2), telemetry (3), camera (4), LEDs (5), depressor (6), during deployment from R/V POSEIDON in February 2018.

Figure 2: Stairwise trajectory of PELAGIOS through the water column, to the desired depths with
concomitantly measured environmental data.

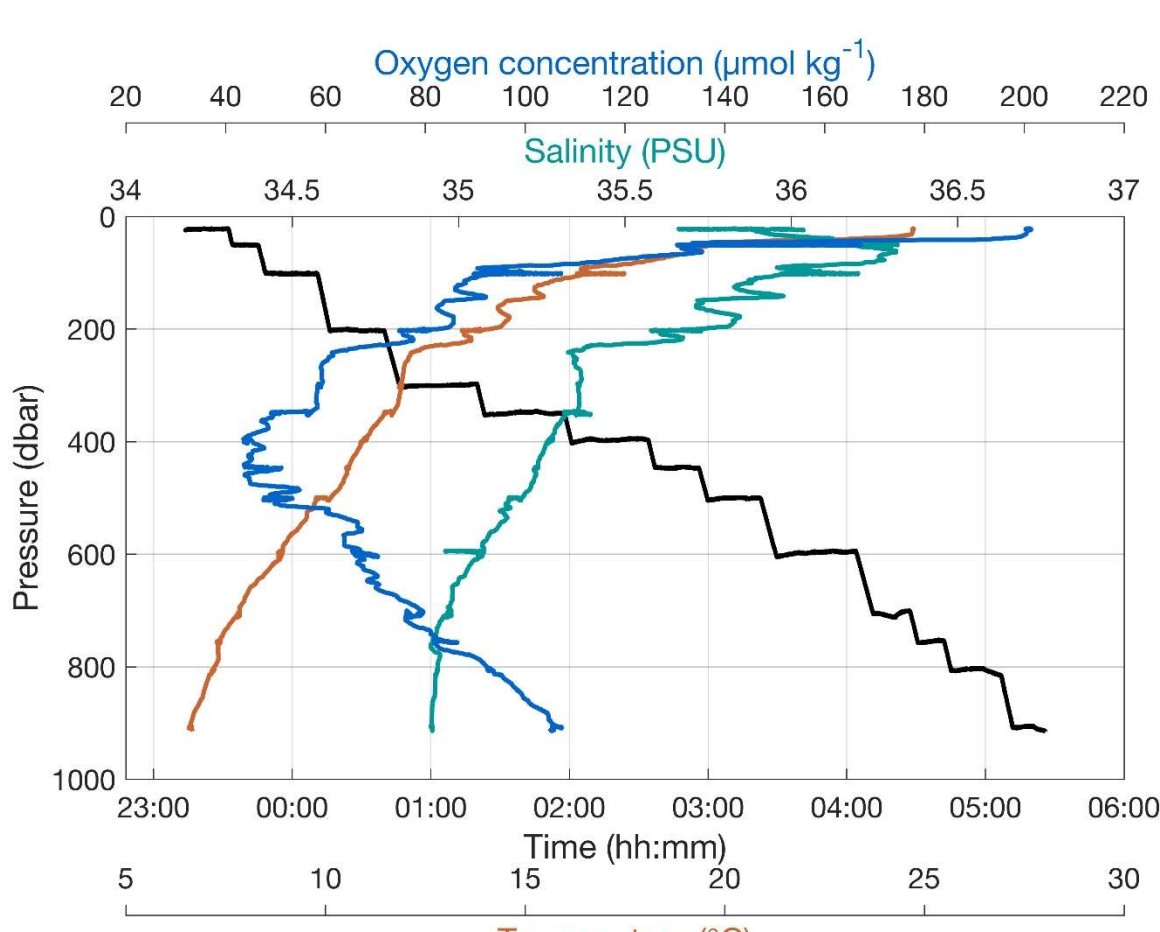








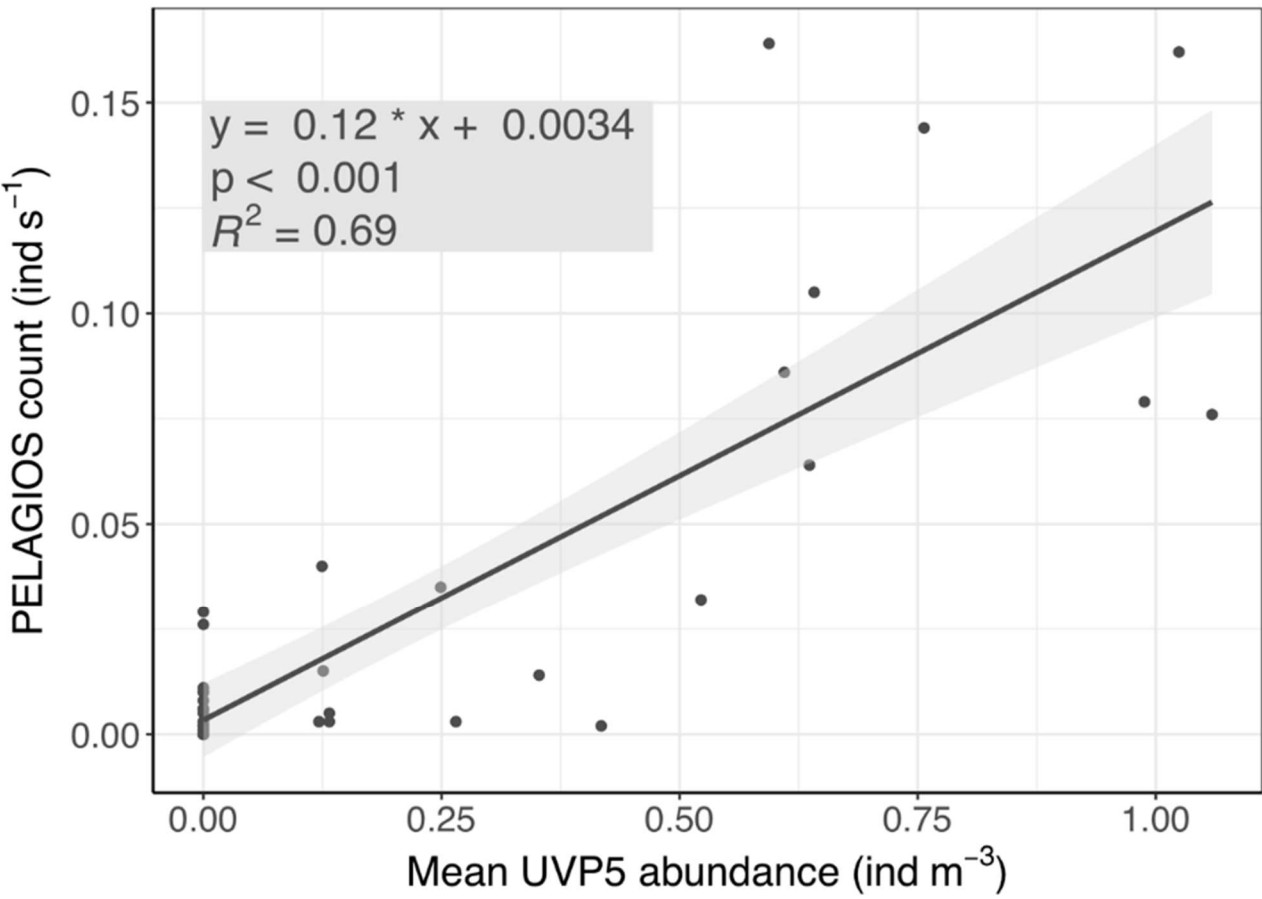



Figure 3: PELAGIOS video counts of *Poeobius* sp. as a function of UVP5-derived abundance on
the same transects at two stations on cruise MSM 49 on RV MARIA S. MERIAN.





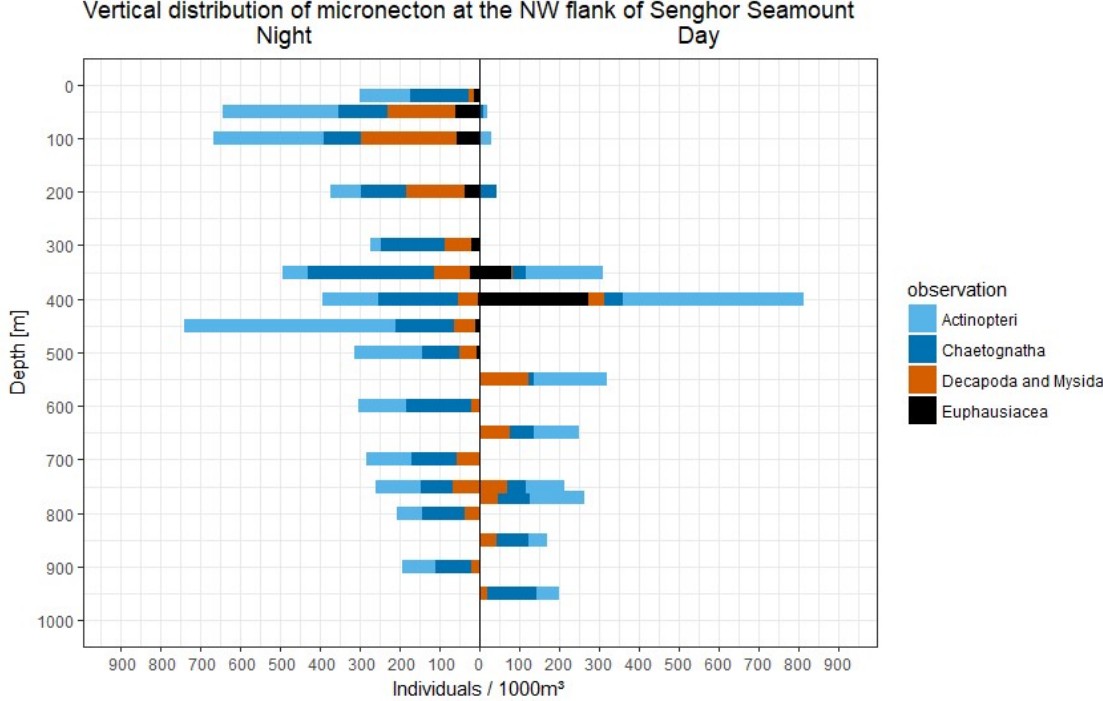


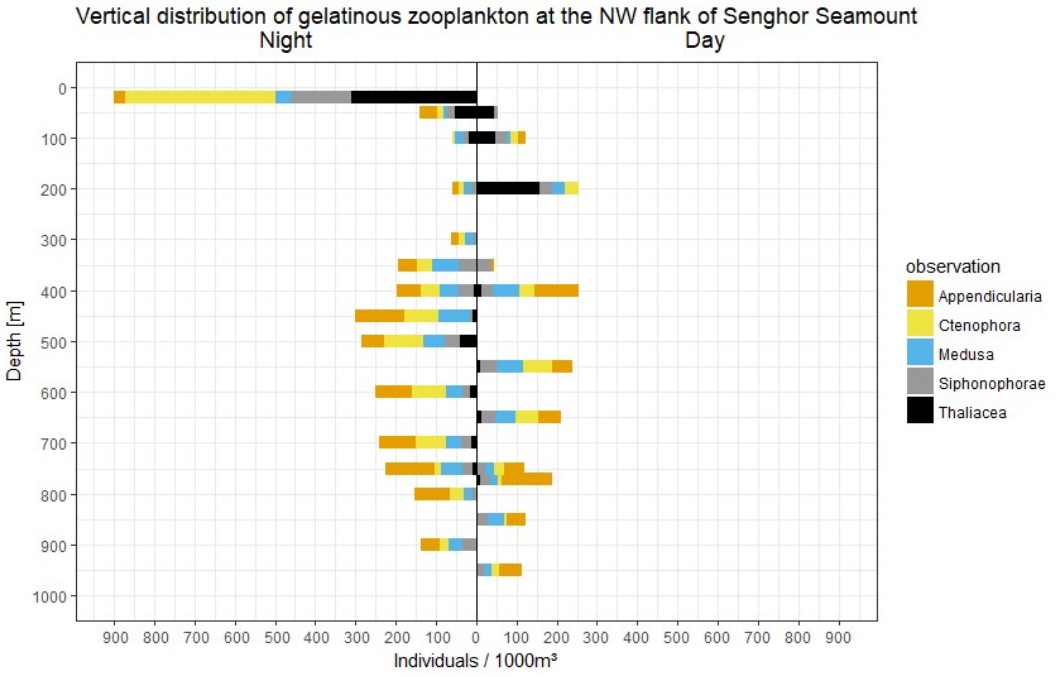


Figure 4: Day and night comparison of faunal observations obtained by PELAGIOS at the North

West flank of Senghor seamount A: fishes, krill, chaetognaths and decapods B: gelatinous

zooplankton groups



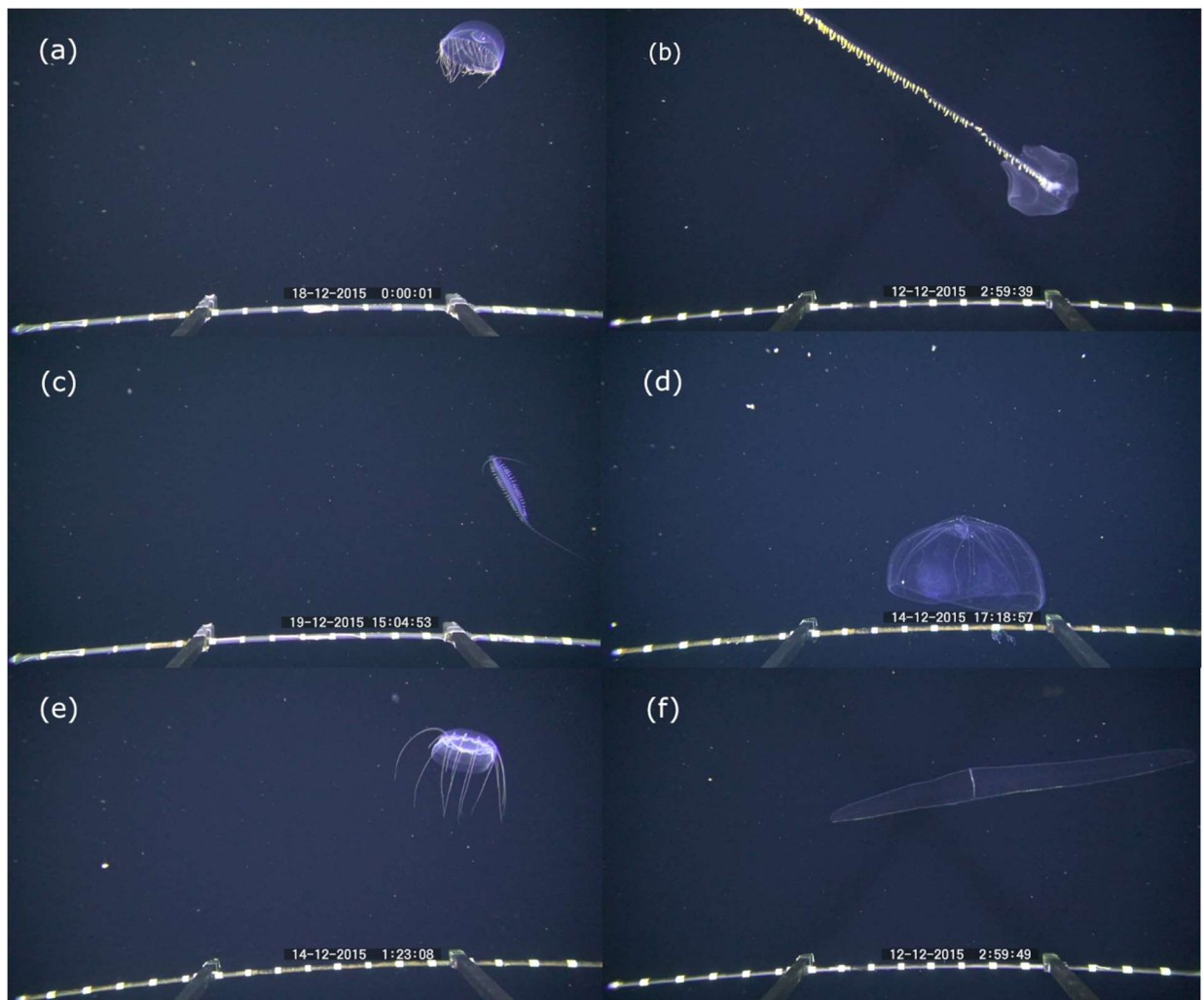

Figure 5: Examples of organisms encountered during pelagic video transects with PELAGIOS during cruise MSM49 in the eastern tropical Atlantic. (a) a medusa *Halitrephes* sp. (b) a siphonophore *Praya dubia* (c) a tomopterid worm (d) the ctenophore *Thalassocalyce inconstans* (e) the medusa *Solmissus* (f) the ctenophore *Cestum*. The distance between the white bands on the horizontal bar on the bottom of the images is 5 cm.


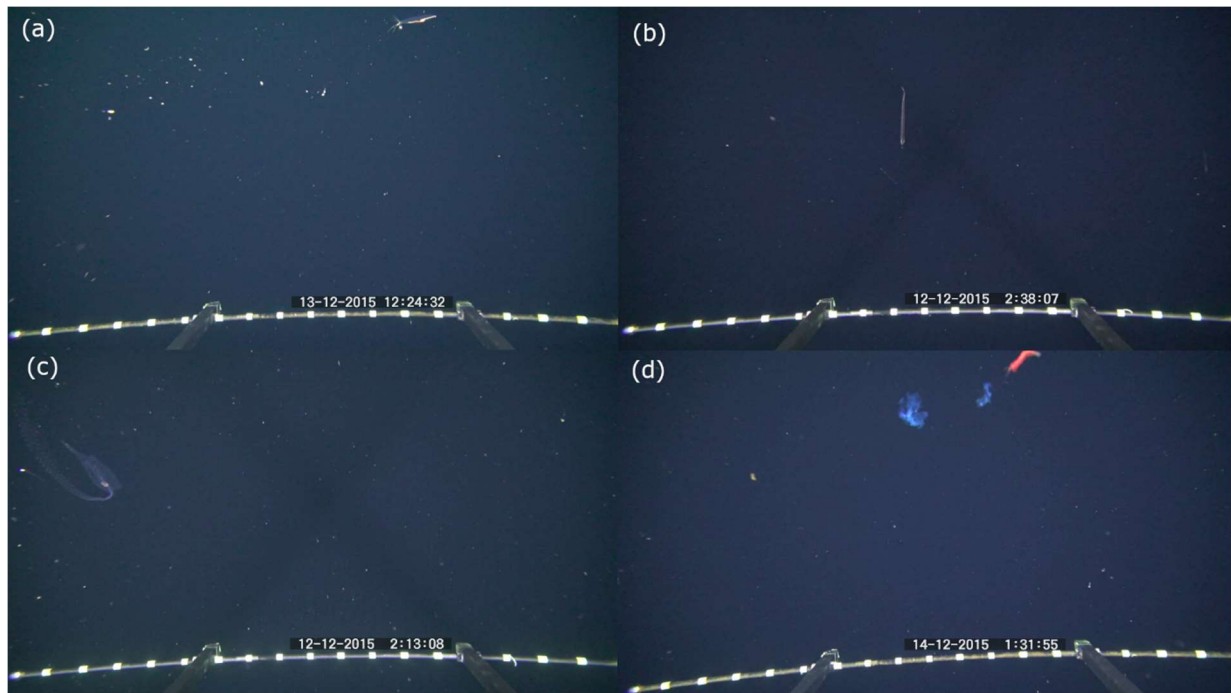


Figure 6: Examples of behaviors observed during pelagic video transects with the PELAGIOS. (a)
*Poeobius* sp. in a feeding position with a mucus web (left side of the animal), (b) a dragonfish of
the family Stomiidae in a vertical position, (c) a salp releasing a blastozoid chain, (d) a crustacean
releasing two bioluminescent clouds while performing an escape response. The distance between
the white bands on the horizontal bar on the bottom of the images is 5 cm.



Table 1: Taxonomic groups which were encountered during pelagic video transects in the eastern tropical Atlantic.

| Phylum | Class | Order | Family | Genus |
|---|---|---|---|---|
| Cercozoa | Thecofilosea | | | |
| Radiozoa | | | | |
| Cnidaria | Hydrozoa | Narcomedusae | Solmundaeginidae | *Solmundella* |
| | | | Aeginidae | *Aegina* |
| | | | | *Aeginura* |
| | | | Cuninidae | *Solmissus* |
| | | Trachymedusae | Halicreatidae | *Halicreas* |
| | | | | *Haliscera* |
| | | | | *Halitrephes* |
| | | | Rhopalonematidae | *Colobonema* |
| | | | | *Crossota* |
| | | | | *Rhopalonema* |
| | | | Geryoniidae | *Geryonia* |
| | | | | *Liriope* |
| | | Siphonophorae | Agalmatidae | *Halistemma* |
| | | | | *Marrus* |
| | | | | *Nanomia* |
| | | | Apolemiidae | *Apolemia* |
| | | | Diphyidae | |
| | | | Forskaliidae | *Forskalia* |
| | | | Hippopodiidae | *Hippopodius* |
| | | | | *Vogtia* |
| | | | Physophoridae | *Physophora* |
| | | | Prayidae | *Craseoa* |
| | | | | *Lilyopsis* |
| | | | | *Praya* |
| | | | | *Rosacea* |
| | | | Pyrostephidae | *Bargmannia* |
| | | | Resomiidae | *Resomia* |
| | Scyphozoa | Coronatae | Atollidae | *Atolla* |
| | | | Nausithoidae | *Nausithoe* |
| | | | Peryphyllidae | *Periphylla* |
| Ctenophora | Nuda | Beroida | Beroidae | *Beroe* |
| | Tentaculata | Cestida | Cestidae | *Cestum* |
| | | | | *Velamen* |
| | | Cydippida | Aulacoctenidae | *Aulacoctena* |
| | | | Pleurobrachiidae | *Hormiphora* |
| | | Lobata | Bathocyroidae | *Bathocyroe* |
| | | | Eurhamphaeidae | *Kiyohimea* |
| | | | Leucotheidae | *Leucothea* |
| | | | Ocryopsidae | *Ocyropsis* |
| | | Thalassocalycida | Thalassocalycidae | *Thalassocalyce* |

| | | | | |
|---|---|---|---|---|
| Chaeotognatha | Sagittoidea | | | |
| Annelida | Polychaeta | Phyllodocida Canalipalpata | Tomopteridae Flabelligeridae | *Tomopteris* *Poeobius* |
| Arthropoda | Malacostraca | Amphipoda Decapoda Euphausiacea Isopoda | Munnopsidae | *Munnopsis* |
| Mollusca | Cephalopoda | Octopoda Teuthida | Amphitretidae Octopodidae Cranchiidae Mastigoteuthidae Octopoteuthidae Ommastrephidae | *Helicocranchia* *Mastigoteuthis* *Octopoteuthis* *Taningia* *Sthenoteuthis* |
| | Gastropoda | Nudibranchia Pteropoda | Phylliroidae | *Phylliroe* |
| Chordata | Appendicularia | Copelata | Oikopleuridae | *Bathochordaeus* *Mesochordaeus* |
| | Thaliacea | Doliolida Pyrosomatida Salpida | Pyrosomatidae Salpidae | *Pyrostemma* *Cyclosalpa* |
| | Actinopteri | Anguilliformes Myctophiformes Stomiiformes | Nemichthyidae Myctophidae Gonostomatidae Sternoptychidae | *Cyclothone* |

