# Peer review of "The Pelagic In situ Observation System (PELAGIOS) to reveal"

_Ocean Science, 2018_

## Referee Comment (RC1) · Anonymous Referee #1 · 3 Jan 2019

Review of the Ocean Science Ms. No. os-2018-131 The Pelagic In situ Observation System (PELAGIOS) to reveal biodiversity, behavior and ecology of elusive oceanic fauna Authors: Hoving et al.

This manuscript provided interesting results but it still needs revisions to be acceptable for publication. To improve the quality and readability of this paper, the following remarks and suggestions are to be considered in view:

Abstract: This part is fine and there is no real need for corrections.

Introduction: Line 32: "have been sampled with nets". You might want to add a reference (e.g., Wiebe and Benfield (2003): From the Hensen net toward four-dimensional

biological oceanography)

Line 33: "a community typically consisting (. . .)" Add a reference.

Sentence at lines 38-42: "This was particularly true for fragile gelatinous zooplankton. . ." add some references. . .

Line 49-50: "pelagic ROV surveys have been applied to study inter and intra-annual variation in mesopelagic zooplankton communities". You can add the following reference: "Hull et al. (2011) Seasonality and depth distribution of a mesopelagic foraminifer, Hastigerinella digitata, in Monterey Bay, California"

Lines 56-60: I would move the Benfield reference to the first sentence.

Line 60: "Examples of instruments include. . ." You can add the following reference to the Zooglider, an in situ imaging device mounted on a glider (something new compared to the other systems you mention). Reference: Ohman et al. (2018?) Zooglider: An autonomous vehicle for optical and acoustic sensing of zooplankton

Material and Method:

Link at line 123 not working. . .

Sub-section 3.4. I am somehow concerned with the way you convert counts/sec to abundances. Are Poebius abundant enough for this kind of comparison? How do deal with patchiness in this comparison? The regression that you show in Figure 3 show multiple points where no Poebius were detected with the UVP, while observed with the Pelagios? How do you explain this discrepancy? If you remove those points, do you still have a significant regression? Is there another way to estimate the Pelagios sampled volume, independently from the UVP comparison? It is important to make this point crystal clear as you are making a direct comparison with MOCNESS abundance later on. . .

Results:

Line 203-223: Do you need to mention every organism that you encountered? Can you somehow make it shorter? It would be nice to have an illustration of the dominant taxa observed by the device (rather than a simple table). It will provide more information for the reader, and potentially raise interest on your device. If you are limited by the number of figures, it could be a supplementary figure...

Line 214: "typical examples of organisms that cannot be captured by nets". Do you have proof of that? (i.e., publication).

Line 214: "can be properly quantified by PELAGIOS". Since you don't have a baseline for your quantification, you cannot say that your device "properly" quantifies these organisms. You might actually undersampled them by having a small sampling volume. You can just say "efficiently observed".

Line 224-233: Refer to my comment for the Methods section... Everything relies here on your conversion factor... A slight change will affect your abundance estimations and ultimately the comparison with MOCNESS abundances... Also, you say that there is an underestimation by MOCNESS but don't provide any data/proof to the reader. Can you summarize the information in a table/figure? Also, why only mentioning the example of Beroe? What about the other taxa mentioned previously (e.g., Poebius?). What's the rationale behind the choice of Beroe?

Sub-section 3.6: Since you made these observations, can you modify Figure 5 (or create a new figure) to provide the visual proof of what you mention in this paragraph?

Discussion:

A general comment regarding this section. There is a lack of references throughout the discussion. We cannot rely only on the author's sayings. I recommend reviewing this section to have clear reference for every/most points you make. Several points are highlighted below.

Line 250: "tool that fills a gap in the array of observation instruments that exist". How

does the PELAGIOS fill a gap? What gap? You have to develop your point here. Viewed from a pessimistic point of view, PELAGIOS can appear as another device wanted by an institution locally, but it will probably never be used outside of this institution. For example, in your introduction, you made the comparison with ROV-video transects. In this case the PELAGIOS appears like an interesting "cost-effective" alternative. Compared to other "well-known" in situ imaging systems (e.g., UVP, VPR), the PELAGIOS does not really provide anything new... You have to better make your point.

Lines 255-257: "The data obtained after annotation of the video can be uploaded into databases (e.g., Pangaea) after publication of the results allowing for efficient data sharing and curation". Any journal requests open-access to published data, you don't have to write this down... Actually, some open-access alternative offers data sharing before publication... (e.g., Ecotaxa, Plankton portal), so it is not even attractive to write such a sentence....

Line 273: "lateral migration of animals towards Senghor seamount at night". Reference?

Line 279: "After annotation, the PELAGIOS video transects may be used to reconstruct species-specific distribution patterns, which can be related to environmental gradients". You have to keep in mind that your device does not provide proper vertical profiles but rather multiple horizontal transects. Compared to other systems (e.g., ISIIS, UVP, VPR, etc.) it does not seem to be the best choice of tool to reconstruct species-specific distribution patterns... You should stress and discuss this point. . .

Line 294: "Preliminary comparisons of the data obtained with PELAGIOS and with MOCNESS indicate substantial differences in the documented fauna". See my comments previously. . . If you don't have further arguments for a robust comparison, you definitely have to stress the uncertainties of your regression. . .

Lines 294-306: Not a single reference here. . . You should include more references in order to provide background information for your argumentation. For example, you did

not mentioned Remsen et al. (2004) paper where similar comparison between imaging device and nets were made...

Lines 307-326: I agree with your point that in situ imaging systems can provide useful information for the significance of fragile organisms to pelagic ecosystems & biogeochemical cycles, but your last comparison with the UVP highlights one of the weakness of the PELAGIOS device. Systems like the UVP or the VPR are not the most advanced systems by far... but they have extensive datasets (like you show). It would take decade for a new system like the PELAGIOS before providing extensive datasets enabling studies a large/global scales. You

Lines 317-320: "This was illustrated by the discovery of the pelagic polychaete Poeobius sp. during the PELAGIOS video transects in the eastern Atlantic (Christiansen et al., 2018). The observations of the PELAGIOS provided the first evidence for the occurrence of Poeobius sp. in the Atlantic Ocean". Isn't the Christiansen paper about UVP data? So, does PELAGIOS provide the first evidence of Poebius in the NA? Also, you then mention the distribution patterns of Poebius, revealed by UVP/CTD and not PELAGIOS... what did PELAGIOS brought to this study (apart from the "discovery"?). If you did not have the UVP/CTD system, would PELAGIOS have been able to provide such information?

Line 330: "The joint deployment of the PELAGIOS and UVP also allowed a quantification of the sampled water volume of the PELAGIOS as described above". See my comments above...

---

## Referee Comment (RC2) · Anonymous Referee #2 · 31 Jan 2019

First, I would like to mention that I'm not an expert in this field and can therefore not comment on the methods. I'm specifically thinking of section 3.4.

While I think the manuscript was carefully written, I did find a few things that need to be clarified.

Lines 56-57 say: "In the last decades, a variety of optical instruments has been developed to image and quantify plankton in situ."

But then lines 73 -75 say: "However, published descriptions of optical systems, other than ROVs and submersibles, that visualize macrozooplankton and micronekton (>1 cm) in the water column are, to the best of our knowledge, restricted to one (Madin et

al., 2006)."

This is confusing as it is currently not clear what the difference is between the above mentioned instruments and the ones that have not been described in publications... Maybe mention in lines 73-73 that there are no other instruments capable of capturing such large organisms?

75 ff Please be more specific about what makes PELAGIOS different from LAPIS

Line 123: Link does not work

Line 195: What was the total transect time during the night? Must be the same amount as during the day, if not, did you account for this in your analysis?

Section 3.5 I find it difficult to read through this section. While it is def. useful to know who lives there, I wonder if there would be a better way to summarize it all in a table and make this section shorter?

213-215: Do you have a reference for this statement?

Minor edits Figure captions Figure 2: Why is O2 plotted but never mentioned? Figure 5: Capitalize "Example"

---

## Author Comment (AC1) · 30 May 2019

This manuscript provided interesting results but it still needs revisions to be acceptable for publication. To improve the quality and readability of this paper, the following remarks and suggestions are to be considered in view:

Abstract: This part is fine and there is no real need for corrections.

Introduction: Line 32: "have been sampled with nets". You might want to add a reference (e.g., Wiebe and Benfield (2003): From the Hensen net toward four-dimensional biological oceanography)

[Figure]

Hoving et al: We added the suggested reference.

Line 33: "a community typically consisting (: : :)" Add a reference.

Hoving et al: We added Benfield et al. 1996 as a reference (comparison MOCNESS to VPR).

Sentence at lines 38-42: "This was particularly true for fragile gelatinous zooplankton.." add some references

Hoving et al: We added and re-organized references to assign references to different delicate faunal groups.

Line 49-50: "pelagic ROV surveys have been applied to study inter and intra-annual variation in mesopelagic zooplankton communities". You can add the following reference: "Hull et al. (2011) Seasonality and depth distribution of a mesopelagic foraminifer, Hastigerinella digitata, in Monterey Bay, California"

Hoving et al: We added suggested reference.

Lines 56-60: I would move the Benfield reference to the first sentence.

Hoving et al: This was moved as suggested.

Line 60: "Examples of instruments include:" You can add the following reference to the Zooglider, an in situ imaging device mounted on a glider (something new compared to the other systems you mention). Reference: Ohman et al. (2018?) Zooglider: An autonomous vehicle for optical and acoustic sensing of zooplankton

Hoving et al: Added suggested reference.

Material and Method: Link at line 123 not working:

Hoving et al: The video has been included as ESM as part of the MS

Sub-section 3.4. I am somehow concerned with the way you convert counts/sec to abundances.

Hoving et al: we have split the questions/concerns and address them separately below.

Are Poebius abundant enough for this kind of comparison?

Hoving et al. We specifically chose Poeobius because its abundance ranged from zero to a (given its size) very high abundance of >1 m-3 . There is no other species that is as abundant and well identifiable in both instruments and that lacks an escape response.

How do deal with patchiness in this comparison?

Hoving et al: For the sake of the regression, we disregard patchiness as we use the mean abundance (ind m-3) and mean count (ind s-1) encountered during an entire transect (between 9 and 22 min).

The regression that you show in Figure 3 show multiple points where no Poebius were detected with the UVP, while observed with the Pelagios? How do you explain this discrepancy? If you remove those points, do you still have a significant regression?

Hoving et al: The sampling volume is much smaller in the UVP, and it does not record continuous video, but image "slices" with a space in between images. This explains the fact that at low abundances Poeobius may be encountered with PELAGIOS, but not imaged by the UVP. If these points are removed, the regression is still significant and the slope changes from 0.12 to 0.13 (see figure A and B attached). The coefficient of determination decreases from 0.69 to 0.52. In our view, it does not make sense to exclude the "zero" observations from the UVP and/or to force the regression through the offspring, because this offset reflects the "missing" Poeobius that are not observed by the UVP at low abundances.

Regression including "zero" observations in the UVP (Figure A) and with these points excluded (Figure B).

Is there another way to estimate the Pelagios sampled volume, independently from the UVP comparison? It is important to make this point crystal clear as you are making a direct comparison with MOCNESS abundance later on.

Hoving et al: One of the future goals is to improve the quantification of the sampled volume, for example by using a current meter. We consider the UVP comparison a good comparison but another way of estimating the field of view is by measuring the area of the image with the scale bar at 1 m from the camera. We inserted this in the text "A cross-sectional view field of approximately 0.23 m2 of PELAGIOS can be expected, compared to a theoretical FOV of 0.45 m2 based upon the maximum image dimensions (0.80 m * 0.56 m) at 1 m distance from the lens." The actual width of view (and hence the field of view) is likely less wide since the view deteriorates to the side. We have moved the PELAGIOS and MOCNESS comparison to the discussion.

Results: Line 203-223: Do you need to mention every organism that you encountered? Can you somehow make it shorter? It would be nice to have an illustration of the dominant taxa observed by the device (rather than a simple table). It will provide more information for the reader, and potentially raise interest on your device. If you are limited by the number of figures, it could be a supplementary figure.

Hoving et al: We have rewritten this paragraph to be more concise. We have added a figure as suggested, and now have one figure with example gelatinous fauna (Figure 5) and another with observed behaviours (Figure 6).

Line 214: "typical examples of organisms that cannot be captured by nets". Do you have proof of that? (i.e., publication).

Hoving et al: We have changed this sentence to read: "Typical examples of fragile organisms that were not present or identifiable in the MOCNESS samples but which can be efficiently observed by PELAGIOS include (. . .)" to clarify that we here directly refer to comparative net hauls (specified before as we moved the MOCNESS comparison up).

Line 214: "can be properly quantified by PELAGIOS". Since you don't have a baseline for your quantification, you cannot say that your device "properly" quantifies these organisms. You might actually undersampled them by having a small sampling volume.

You can just say "efficiently observed".

Hoving et al: changed according to suggestion.

Line 224-233: Refer to my comment for the Methods section. Everything relies here on your conversion factor. A slight change will affect your abundance estimations and ultimately the comparison with MOCNESS abundances. Also, you say that there is an underestimation by MOCNESS but don't provide any data/proof to the reader. Can you summarize the information in a table/figure? Also, why only mentioning the example of Beroe? What about the other taxa mentioned previously (e.g., Poebius?). What's the rationale behind the choice of Beroe?

Hoving et al: For intercomparison between two instruments, we need to choose organisms that we can identify in both. Beroe is an example of a comparatively large, sturdy ctenophore that could also be identified in net hauls, but seems to be underestimated as is it often severed in the catch. As for Poeobius, we have never been able to retrieve this organism using nets in the Eastern Tropical Atlantic (not even with a small $200\mu$m multinet), but we can identify it on UVP images, and since it does not have an escape response and falls well in the UVP size range, we assume that UVP observations are quantitative. We have added some considerations on the accuracy of the sampling volume and area in the results and moved the comparison with MOCNESS to the discussion (lines 379-391).

Sub-section 3.6: Since you made these observations, can you modify Figure 5 (or create a new figure) to provide the visual proof of what you mention in this paragraph?

Hoving et al: We have added a new figure (Figure 6) that illustrates the behaviours observed with PELAGIOS as described in the text.

Discussion: A general comment regarding this section. There is a lack of references throughout the discussion. We cannot rely only on the author's sayings. I recommend reviewing this section to have clear reference for every/most points you make. Several

points are highlighted below. Line 250: "tool that fills a gap in the array of observation instruments that exist". How does the PELAGIOS fill a gap? What gap? You have to develop your point here.

Hoving et al: We have added a couple of sentences to better clarify the need for video observations on transparent, fragile fauna (lines 88-93; 295-306). We also added additional references.

Viewed from a pessimistic point of view, PELAGIOS can appear as another device wanted by an institution locally, but it will probably never be used outside of this institution. For example, in your introduction, you made the comparison with ROV-video transects. In this case the PELAGIOS appears like an interesting "cost-effective" alternative. Compared to other "well-known" in situ imaging systems (e.g., UVP, VPR), the PELAGIOS does not really provide anything new... You have to better make your point.

Hoving et al: PELAGIOS does provide something new. It allows cost effective observations in a similar way as ROV horizontal transects. It allows the visualization of fauna > 1cm. We have tried to better make our point in the first paragraph of the discussion. PELAGIOS does not cover the same range of planktonic organisms that the VPR or UVP do; there is only a fairly small overlap. We are not aware of a functional instrument that does. We do not attempt to compete with the UVP5 but consider them as complementary tools as we show in our comparison.

Lines 255-257: "The data obtained after annotation of the video can be uploaded into databases (e.g., Pangaea) after publication of the results allowing for efficient data sharing and curation". Any journal requests open-access to published data, you don't have to write this down... Actually, some open-access alternative offers data sharing before publication... (e.g., Ecotaxa, Plankton portal), so it is not even attractive to write such a sentence....

Hoving et al: We have had trouble to obtain raw data from other optical instruments for cross-comparison, so we feel it is valid to point out that data shall be made available

on queriable databases (prior to or after publication).

Line 273: "lateral migration of animals towards Senghor seamount at night". Reference?

Hoving et al: We have changed the sentence and added three references.

Line 279: "After annotation, the PELAGIOS video transects may be used to reconstruct species-specific distribution patterns, which can be related to environmental gradients". You have to keep in mind that your device does not provide proper vertical profiles but rather multiple horizontal transects. Compared to other systems (e.g., ISIIS, UVP, VPR, etc.) it does not seem to be the best choice of tool to reconstruct species-specific distribution patterns... You should stress and discuss this point.

Hoving et al: The PELAGIOS is suitable for visualizing plankton and nekton > 1 cm and therefore is not comparable to ISIIS or UVP and we do not attempt to compete with these devices which are highly suitable for quantification of distribution of meso-zooplankton and particles. The PELAGIOS video transects are comparable to horizontal ROV transects, and can be used to detect fragile fauna and reconstruct species-specific distribution patterns of larger macrozooplankton, as we show here and in cited publications that use PELAGIOS data. Our deployments were so far typically horizontally since we wanted to have more data from one depth to reconstruct the vertical species distributions. If desired one could deploy PELAGIOS vertically for studies on spatial distribution.

Line 294: "Preliminary comparisons of the data obtained with PELAGIOS and with MOCNESS indicate substantial differences in the documented fauna". See my comments previously. If you don't have further arguments for a robust comparison, you definitely have to stress the uncertainties of your regression.

Hoving et al: We have moved the section on the comparison between PELAGIOS and MOCNESS to the discussion section to emphasize it is an exploration of the obtained

data. We particularly refer to the difference in number of taxa in this paragraph, and explore the quantitative difference using the volume from the UVP-PELAGIOS comparison. The uncertainty of this regression is given in the manuscript. Even without the quantitative comparison, and considering only the presence and absence data, substantial differences are obvious. We also state that we are striving to improve the quantitative sampling of the system as part of future development.

Lines 294-306: Not a single reference here. You should include more references in order to provide background information for your argumentation. For example, you did not mentioned Remsen et al. (2004) paper where similar comparison between imaging device and nets were made.

Hoving et al: We have added more references throughout the discussion including Remsen et al 2004

Lines 307-326: I agree with your point that in situ imaging systems can provide useful information for the significance of fragile organisms to pelagic ecosystems & biogeochemical cycles, but your last comparison with the UVP highlights one of the weakness of the PELAGIOS device. Systems like the UVP or the VPR are not the most advanced systems by far but they have extensive datasets (like you show). It would take decade for a new system like the PELAGIOS before providing extensive datasets enabling studies a large/global scales.

Hoving et al: Even if PELAGIOS does not turn out a standard observation instrument (such as the UVP and VPR, which can be readily integrated to other platforms and have a streamlined image processing pipeline), it is a valuable tool to quantify organisms that are up to now missed by any other quantitative routine observational system, and that are play important roles in the ecosystem and for biogeochemical cycles. We have added sentences in the first paragraph of the discussion to point out where the instrument fills a gap. At the same time, PELAGIOS can be adapted to fit on a CTD or other plankton observation platforms, and with enough effort, large datasets can follow.

It should again be mentioned that PELAGIOS collects video transect data and has a different purpose that the UVP and VPR. See earlier comments.

Lines 317-320: "This was illustrated by the discovery of the pelagic polychaete Poeobius sp. during the PELAGIOS video transects in the eastern Atlantic (Christiansen et al., 2018). The observations of the PELAGIOS provided the first evidence for the occurrence of Poeobius sp. in the Atlantic Ocean". Isn't the Christiansen paper about UVP data? So, does PELAGIOS provide the first evidence of Poebius in the NA? Also, you then mention the distribution patterns of Poebius, revealed by UVP/CTD and not PELAGIOS? what did PELAGIOS brought to this study (apart from the "discovery"?). If you did not have the UVP/CTD system, would PELAGIOS have been able to provide such information?

Hoving et al: Yes, PELAGIOS did provide the first video observation of Poeobius in the Atlantic. Only after this discovery, we checked the extensive UVP image database, found it there as well and created a category for automatic sorting (followed by manual validation) for all available profiles, which then resulted in the dataset presented in Christiansen et al. 2018. The PELAGIOS also provided in situ observations that allowed the estimation of the size of the mucus net for the study. While most of the distribution data came from the UVP5, the discovery was made by PELAGIOS. It was the combination of tools that made an integrative detailed study on the ecology of the species possible.

Line 330: "The joint deployment of the PELAGIOS and UVP also allowed a quantification of the sampled water volume of the PELAGIOS as described above". See my comments above.

Hoving et al: comments noted and responded to
* * *
**Fig. 1.**

[Figure]

$$y = 0.13 * x + -0.0051$$
$$p < 0.001$$
$$R^2 = 0.52$$

PELAGIOS count (ind s$^{-1}$)

Mean UVP5 abundance (ind m$^{-3}$)

**Fig. 2.**

---

## Author Comment (AC2) · 30 May 2019

Dear Editor, Below follows a response to the comments provided by Referee #2. We list the comment of the referee and respond to it after 'Hoving et al'.

Referee #2: First, I would like to mention that I'm not an expert in this field and can therefore not comment on the methods. I'm specifically thinking of section 3.4. While I think the manuscript was carefully written, I did find a few things that need to be clarified. Lines 56-57 say: "In the last decades, a variety of optical instruments has been developed to image and quantify plankton in situ." But then lines 73 -75 say: "However, published descriptions of optical systems, other than ROVs and submersibles, that vi-

sualize macrozooplankton and micronekton (>1 cm) in the water column are, to the best of our knowledge, restricted to one (Madin et al., 2006)." This is confusing as it is currently not clear what the difference is between the above mentioned instruments and the ones that have not been described in publications. Maybe mention in lines 73-73 that there are no other instruments capable of capturing such large organisms?

Hoving et al: In the revised version, we have tried to point out the novelty and differentiating characteristics of the instrument and that PELAGIOS is mainly designed to make video observations of large, transparent, fragile organisms, which fills a gap in the current instrument array available.

Referee #2: 75 ff Please be more specific about what makes PELAGIOS different from LAPIS

Hoving et al: We have added information about LAPIS that indicates the difference. For example LAPIS used still imagery, PELAGIOS uses video allowing documentation of behaviour; LAPIS has an illuminated box in which the organisms are photographed, PELAGIOS has forward illumination similar to an ROV. PELAGIOS data can be compared with ROV video transects. There are no additional publications that show LAPIS data and hence the development and application of PELAGIOS is timely.

Referee #2: Link at line 123 not working:

Hoving et al: The video will be available as ESM in the MS

Referee #2: Line 195: What was the total transect time during the night? Must be the same amount as during the day, if not, did you account for this in your analysis?

Hoving et al: We included the transect time and corrected the comparison since the night transects were in total longer. In Figure 4 the data are corrected for time.

Referee #2: Section 3.5 I find it difficult to read through this section. While it is def. useful to know who lives there, I wonder if there would be a better way to summarize it all in a table and make this section shorter?

Hoving et al: We have rewritten this paragraph to be more concise and improve readability.

Referee #2: 213-215: Do you have a reference for this statement?

Hoving et al: We have added Harbison et al. 1978 as a reference here.

Referee #2: Minor edits Figure captions Figure 2: Why is O2 plotted but never mentioned? Figure 5: Capitalize "Example" Hoving et al: We have integrated the other sensor data in this figure, as an illustration of complementary video and environmental sensor data collection.

———————————————————

---

## Author Comment (AC3) · 30 May 2019

Dear Editor, Below follows a response to the comments provided by Referee #1. We list the comment of the referee and respond to it after 'Hoving et al'.

Referee #1: This manuscript provided interesting results but it still needs revisions to be acceptable for publication. To improve the quality and readability of this paper, the following remarks and suggestions are to be considered in view:

Referee #1: Abstract: This part is fine and there is no real need for corrections.

Referee #1: Introduction: Line 32: "have been sampled with nets". You might want

to add a reference (e.g., Wiebe and Benfield (2003): From the Hensen net toward four-dimensional biological oceanography)

Hoving et al: We added the suggested reference.

Referee #1: Line 33: "a community typically consisting (: : :)" Add a reference.

Hoving et al: We added Benfield et al. 1996 as a reference (comparison MOCNESS to VPR).

Referee #1: Sentence at lines 38-42: "This was particularly true for fragile gelatinous zooplankton.." add some references

Hoving et al: We added and re-organized references to assign references to different delicate faunal groups.

Referee #1: Line 49-50: "pelagic ROV surveys have been applied to study inter and intra-annual variation in mesopelagic zooplankton communities". You can add the following reference: "Hull et al. (2011) Seasonality and depth distribution of a mesopelagic foraminifer, Hastigerinella digitata, in Monterey Bay, California"

Hoving et al: We added suggested reference.

Referee #1: Lines 56-60: I would move the Benfield reference to the first sentence.

Hoving et al: This was moved as suggested.

Referee #1: Line 60: "Examples of instruments include:" You can add the following reference to the Zooglider, an in situ imaging device mounted on a glider (something new compared to the other systems you mention). Reference: Ohman et al. (2018?) Zooglider: An autonomous vehicle for optical and acoustic sensing of zooplankton

Hoving et al: Added suggested reference.

Referee #1: Material and Method: Link at line 123 not working:

Hoving et al: The video has been included as ESM as part of the MS

[Figure]

Referee #1: Sub-section 3.4. I am somehow concerned with the way you convert counts/sec to abundances.

Hoving et al: we have split the questions/concerns and address them separately below.

Referee #1: Are Poebius abundant enough for this kind of comparison?

Hoving et al. We specifically chose Poeobius because its abundance ranged from zero to a (given its size) very high abundance of >1 m-3 . There is no other species that is as abundant and well identifiable in both instruments and that lacks an escape response.

Referee #1: How do deal with patchiness in this comparison?

Hoving et al: For the sake of the regression, we disregard patchiness as we use the mean abundance (ind m-3) and mean count (ind s-1) encountered during an entire transect (between 9 and 22 min).

Referee #1: The regression that you show in Figure 3 show multiple points where no Poebius were detected with the UVP, while observed with the Pelagios? How do you explain this discrepancy? If you remove those points, do you still have a significant regression?

Hoving et al: The sampling volume is much smaller in the UVP, and it does not record continuous video, but image "slices" with a space in between images. This explains the fact that at low abundances Poeobius may be encountered with PELAGIOS, but not imaged by the UVP. If these points are removed, the regression is still significant and the slope changes from 0.12 to 0.13 (see figure A and B attached). The coefficient of determination decreases from 0.69 to 0.52. In our view, it does not make sense to exclude the "zero" observations from the UVP and/or to force the regression through the offspring, because this offset reflects the "missing" Poeobius that are not observed by the UVP at low abundances.

Referee #1: Regression including "zero" observations in the UVP (Figure A) and with these points excluded (Figure B).
Referee #1: Is there another way to estimate the Pelagios sampled volume, independently from the UVP comparison? It is important to make this point crystal clear as you are making a direct comparison with MOCNESS abundance later on.

Hoving et al: One of the future goals is to improve the quantification of the sampled volume, for example by using a current meter. We consider the UVP comparison a good comparison but another way of estimating the field of view is by measuring the area of the image with the scale bar at 1 m from the camera. We inserted this in the text "A cross-sectional view field of approximately 0.23 m2 of PELAGIOS can be expected, compared to a theoretical FOV of 0.45 m2 based upon the maximum image dimensions (0.80 m * 0.56 m) at 1 m distance from the lens." The actual width of view (and hence the field of view) is likely less wide since the view deteriorates to the side. We have moved the PELAGIOS and MOCNESS comparison to the discussion.

Results: Referee #1: Line 203-223: Do you need to mention every organism that you encountered? Can you somehow make it shorter? It would be nice to have an illustration of the dominant taxa observed by the device (rather than a simple table). It will provide more information for the reader, and potentially raise interest on your device. If you are limited by the number of figures, it could be a supplementary figure.

Hoving et al: We have rewritten this paragraph to be more concise. We have added a figure as suggested, and now have one figure with example gelatinous fauna (Figure 5) and another with observed behaviours (Figure 6).

Referee #1: Line 214: "typical examples of organisms that cannot be captured by nets". Do you have proof of that? (i.e., publication).

Hoving et al: We have changed this sentence to read: "Typical examples of fragile organisms that were not present or identifiable in the MOCNESS samples but which can be efficiently observed by PELAGIOS include (...)" to clarify that we here directly refer to comparative net hauls (specified before as we moved the MOCNESS comparison up).

Referee #1: Line 214: "can be properly quantified by PELAGIOS". Since you don't have a baseline for your quantification, you cannot say that your device "properly" quantifies these organisms. You might actually undersampled them by having a small sampling volume. You can just say "efficiently observed".

Hoving et al: changed according to suggestion.

Referee #1: Line 224-233: Refer to my comment for the Methods section. Everything relies here on your conversion factor. A slight change will affect your abundance estimations and ultimately the comparison with MOCNESS abundances. Also, you say that there is an underestimation by MOCNESS but don't provide any data/proof to the reader. Can you summarize the information in a table/figure? Also, why only mentioning the example of Beroe? What about the other taxa mentioned previously (e.g., Poebius?). What's the rationale behind the choice of Beroe?

Hoving et al: For intercomparison between two instruments, we need to choose organisms that we can identify in both. Beroe is an example of a comparatively large, sturdy ctenophore that could also be identified in net hauls, but seems to be underestimated as is it often severed in the catch. As for Poeobius, we have never been able to retrieve this organism using nets in the Eastern Tropical Atlantic (not even with a small $200\mu$m multinet), but we can identify it on UVP images, and since it does not have an escape response and falls well in the UVP size range, we assume that UVP observations are quantitative. We have added some considerations on the accuracy of the sampling volume and area in the results and moved the comparison with MOCNESS to the discussion (lines 379-391).

Referee #1: Sub-section 3.6: Since you made these observations, can you modify Figure 5 (or create a new figure) to provide the visual proof of what you mention in this paragraph?

Hoving et al: We have added a new figure (Figure 6) that illustrates the behaviours observed with PELAGIOS as described in the text.

Discussion: Referee #1: A general comment regarding this section. There is a lack of references throughout the discussion. We cannot rely only on the author's sayings. I recommend reviewing this section to have clear reference for every/most points you make. Several points are highlighted below. Line 250: "tool that fills a gap in the array of observation instruments that exist". How does the PELAGIOS fill a gap? What gap? You have to develop your point here.

Hoving et al: We have added a couple of sentences to better clarify the need for video observations on transparent, fragile fauna (lines 88-93; 295-306). We also added additional references.

Referee #1: Viewed from a pessimistic point of view, PELAGIOS can appear as another device wanted by an institution locally, but it will probably never be used outside of this institution. For example, in your introduction, you made the comparison with ROV-video transects. In this case the PELAGIOS appears like an interesting "cost-effective" alternative. Compared to other "well-known" in situ imaging systems (e.g., UVP, VPR), the PELAGIOS does not really provide anything new... You have to better make your point.

Hoving et al: PELAGIOS does provide something new. It allows cost effective observations in a similar way as ROV horizontal transects. It allows the visualization of fauna > 1cm. We have tried to better make our point in the first paragraph of the discussion. PELAGIOS does not cover the same range of planktonic organisms that the VPR or UVP do; there is only a fairly small overlap. We are not aware of a functional instrument that does. We do not attempt to compete with the UVP5 but consider them as complementary tools as we show in our comparison.

Referee #1: Lines 255-257: "The data obtained after annotation of the video can be uploaded into databases (e.g., Pangaea) after publication of the results allowing for efficient data sharing and curation". Any journal requests open-access to published data, you don't have to write this down... Actually, some open-access alternative offers

data sharing before publication... (e.g., Ecotaxa, Plankton portal), so it is not even attractive to write such a sentence....

Hoving et al: We have had trouble to obtain raw data from other optical instruments for cross-comparison, so we feel it is valid to point out that data shall be made available on queriable databases (prior to or after publication).

Referee #1: Line 273: "lateral migration of animals towards Senghor seamount at night". Reference?

Hoving et al: We have changed the sentence and added three references.

Referee #1: Line 279: "After annotation, the PELAGIOS video transects may be used to reconstruct species-specific distribution patterns, which can be related to environmental gradients". You have to keep in mind that your device does not provide proper vertical profiles but rather multiple horizontal transects. Compared to other systems (e.g., ISIIS, UVP, VPR, etc.) it does not seem to be the best choice of tool to reconstruct species-specific distribution patterns... You should stress and discuss this point.

Hoving et al: The PELAGIOS is suitable for visualizing plankton and nekton > 1 cm and therefore is not comparable to ISIIS or UVP and we do not attempt to compete with these devices which are highly suitable for quantification of distribution of meso-zooplankton and particles. The PELAGIOS video transects are comparable to horizontal ROV transects, and can be used to detect fragile fauna and reconstruct species-specific distribution patterns of larger macrozooplankton, as we show here and in cited publications that use PELAGIOS data. Our deployments were so far typically horizontally since we wanted to have more data from one depth to reconstruct the vertical species distributions. If desired one could deploy PELAGIOS vertically for studies on spatial distribution.

Referee #1: Line 294: "Preliminary comparisons of the data obtained with PELAGIOS and with MOCNESS indicate substantial differences in the documented fauna". See

my comments previously. If you don't have further arguments for a robust comparison, you definitely have to stress the uncertainties of your regression.

Hoving et al: We have moved the section on the comparison between PELAGIOS and MOCNESS to the discussion section to emphasize it is an exploration of the obtained data. We particularly refer to the difference in number of taxa in this paragraph, and explore the quantitative difference using the volume from the UVP-PELAGIOS comparison. The uncertainty of this regression is given in the manuscript. Even without the quantitative comparison, and considering only the presence and absence data, substantial differences are obvious. We also state that we are striving to improve the quantitative sampling of the system as part of future development.

Referee #1: Lines 294-306: Not a single reference here. You should include more references in order to provide background information for your argumentation. For example, you did not mentioned Remsen et al. (2004) paper where similar comparison between imaging device and nets were made.

Hoving et al: We have added more references throughout the discussion including Remsen et al 2004

Referee #1: Lines 307-326: I agree with your point that in situ imaging systems can provide useful information for the significance of fragile organisms to pelagic ecosystems & biogeochemical cycles, but your last comparison with the UVP highlights one of the weakness of the PELAGIOS device. Systems like the UVP or the VPR are not the most advanced systems by far but they have extensive datasets (like you show). It would take decade for a new system like the PELAGIOS before providing extensive datasets enabling studies a large/global scales.

Hoving et al: Even if PELAGIOS does not turn out a standard observation instrument (such as the UVP and VPR, which can be readily integrated to other platforms and have a streamlined image processing pipeline), it is a valuable tool to quantify organisms that are up to now missed by any other quantitative routine observational system, and

that are play important roles in the ecosystem and for biogeochemical cycles. We have added sentences in the first paragraph of the discussion to point out where the instrument fills a gap. At the same time, PELAGIOS can be adapted to fit on a CTD or other plankton observation platforms, and with enough effort, large datasets can follow. It should again be mentioned that PELAGIOS collects video transect data and has a different purpose that the UVP and VPR. See earlier comments.

Referee #1: Lines 317-320: "This was illustrated by the discovery of the pelagic polychaete Poeobius sp. during the PELAGIOS video transects in the eastern Atlantic (Christiansen et al., 2018). The observations of the PELAGIOS provided the first evidence for the occurrence of Poeobius sp. in the Atlantic Ocean". Isn't the Christiansen paper about UVP data? So, does PELAGIOS provide the first evidence of Poebius in the NA? Also, you then mention the distribution patterns of Poebius, revealed by UVP/CTD and not PELAGIOS? what did PELAGIOS brought to this study (apart from the "discovery"?). If you did not have the UVP/CTD system, would PELAGIOS have been able to provide such information?

Hoving et al: Yes, PELAGIOS did provide the first video observation of Poeobius in the Atlantic. Only after this discovery, we checked the extensive UVP image database, found it there as well and created a category for automatic sorting (followed by manual validation) for all available profiles, which then resulted in the dataset presented in Christiansen et al. 2018. The PELAGIOS also provided in situ observations that allowed the estimation of the size of the mucus net for the study. While most of the distribution data came from the UVP5, the discovery was made by PELAGIOS. It was the combination of tools that made an integrative detailed study on the ecology of the species possible.

Referee #1: Line 330: "The joint deployment of the PELAGIOS and UVP also allowed a quantification of the sampled water volume of the PELAGIOS as described above". See my comments above.

[Figure]

Hoving et al: comments noted and responded to

$$y = 0.12 * x + 0.0034$$
$$p < 0.001$$
$$R^2 = 0.69$$

**Fig. 1.**

[Figure]

PELAGIOS count (ind s$^{-1}$)

$y = 0.13 * x + -0.0051$
$p < 0.001$
$R^2 = 0.52$

0.15

0.10

0.05

0.00

0.00      0.25      0.50      0.75      1.00

Mean UVP5 abundance (ind m$^{-3}$)

**Fig. 2.**

---

## Author Response (AR2)

**Dear Editor,**

**Please find below a short response to each of the raised issues.**

**Thank you very much for accepting our paper in Ocean Science.**

**Kind regards Henk-Jan Hoving**

L45 " … the collection of fine-scale distribution patterns …" Distribution patterns of what?

Hoving: we have clarified that it involves gelatinous zooplanktonic organisms

L61 Please define CTD

Hoving: I do not think this is necessary since CTD is a very well known instrument for the scientists who are reading our paper.

L183 I have the impression that the given m/s velocity has too many digitals, since the speed is not that precise.

Hoving: Accepted

L202-203 "and somewhat for fishes" It was not directly clear to me what was meant here. Please modify the sentence.

Hoving: I have removed 'somewhat'

L264-268 This sentence is unclear. Please modify it to make it clear.

Hoving: We have modified this sentence

L269 "The relatively simple design limits technical failures …" Is this an expectation or based on experience?

Hoving: I have indicated that this is based on experience.

L302 (also L307) What do you mean with "environmental gradients". Would environmental conditions also do?

Hoving: we have rewritten this part and included conditions.

L389-402 Maybe this last paragraph could appear under the section header Conclusions. I think a paper generally needs a Conclusion section.

Hoving: We prefer to leave it as it is since it is more of an outlook to future developments than a conclusion.

L416 The datasets generated and/or analysed during the current study ARE available …

Hoving: accepted and changed

References
L457 Choy et al: Please add doi
L459 Please add info on publication. What is it, where published?
L465 Please check ref. It is not complete.
L524 Please check ref. Page numbers are wrong.
L547 Please check ref. It is incomplete
L555 Check ref. Page numbers wrong.
L588 Check ref. It is incomplete

Hoving: all references are now fixed and we have changed one reference about LOKI to enable a reference to a journal